# Visualization and Bibliometric Analysis of Research Trends on Hyperbaric Oxygen Therapy

**DOI:** 10.3390/ijerph19137866

**Published:** 2022-06-27

**Authors:** Nan-Chieh Huang, Yu-Lung Wu, Ren-Fang Chao

**Affiliations:** 1Department of Information Engineering, I-Shou University, Kaohsiung City 84001, Taiwan; pippen6233@yahoo.com.tw; 2Division of Family Medicine, Zuoying Branch of Kaohsiung Armed Forces General Hospital, Kaohsiung City 813204, Taiwan; 3Department of Information Management, I-Shou University, Kaohsiung City 84001, Taiwan; wuyulung@isu.edu.tw; 4Department of Leisure Management, I-Shou University, Kaohsiung City 84001, Taiwan

**Keywords:** hyperbaric oxygen therapy, bibliometrics, VOSviewer visualization analysis

## Abstract

Hyperbaric oxygen therapy (HBOT) is a professional medical regimen with a wide range of clinical applications in various research fields. In addition to treating diving decompression sickness and air embolism, HBOT is used as an adjuvant in the management of various diseases. A large number of studies have been published to confirm its efficacy. Although HBOT has been clinically applied to the treatment of many diseases, the effectiveness of these treatments remains controversial. Exploring and evaluating HBOT will contribute to the future development of research in this field. Through a quantitative analysis of the literature, this paper explores the citation relevance and collaboration map and their impact on research outcomes. This study used bibliometric and cartographic techniques with VOSviewer to identify the most influential countries and scholars using this treatment, based on syndrome differentiation. It also provides continuous quality evaluation and lean management of the medical expenses associated with HBOT.

## 1. Introduction

Hyperbaric oxygen therapy (HBOT) is a recognized treatment for diving decompression sickness (DCS) [1] and air embolism [2]. Hyperbaric oxygen has been used in animal experiments since as early as 1887, when Valenzuela [3] showed that, after injecting rotten matter from dead rabbits into live rabbits, hyperbaric oxygen treatment reduced inflammation and improved the survival rate. Behnke and Shaw [4] began treating DCS with HBOT in 1937. With progress in medical research, the mechanism of action, efficacy, indications, complications and contraindications of HBOT have been gradually verified and confirmed [5]. For example, HBOT can be used as an adjunctive treatment for refractory wounds [6]. In addition, HBOT has therapeutic effects by accelerating the body’s physiological metabolism and improving fatigue after exercise [7]. Conceptually, the repair of damaged tissue is based on an increase in the partial pressure and concentration of oxygen in tissue; this effect can enhance the physiological response of cells and tissue. It is still used to treat necrotising soft tissue infections [8], air or gas embolism, diabetic foot ulcers [9] and radiation damage [10]. It is even used in some diving medical research and rescue operations [11]. The Undersea and Hyperbaric Medical Society (UHMS) lists the indications of HBOT that are reimbursed in hospitals and clinics [12]. The Therapeutic Committee has approved (reimbursed) the following indications: decompression sickness; bubbles of air in blood vessels (air or gas embolism); carbon monoxide poisoning; idiopathic sudden sensorineural hearing loss [13]; myonecrosis; central retinal artery occlusion [14]; ischemia; cyanide poisoning; compartment syndrome; clostridial myositis; diabetically derived illness (such as difficulties in foot wound healing, retinopathy and nephropathy); delayed radiation injury; necrotizing soft osteomyelitis (refractory) [15]; crush injury and other acute trauma; exceptional blood loss (anaemia); enhancement of healing in selected problem wounds; intracranial abscess; mucormycosis; thermal burns (skin grafts and flaps) [16]; and tissue infections (necrotizing fasciitis) [17].

In the past, most of the academic research on HBOT has focused on its treatment mechanisms and therapeutic effects on a disease [18]. However, even though HBOT has a long research history, a comprehensive overview of bibliometrics is still lacking so far. The bibliometric method is used to quantitatively analyse the research literature through database structure calculation and statistics [19]. This analysis can assess the research performance of groups and individuals [20] and identify academic characteristics of exchange and cooperation [21]. Therefore, bibliometric analysis can provide insight into previous studies and thereby directions for future [22]. Its use by academic researchers in various fields, including medicine [23], the humanities, psychology, social sciences [24] and environmental sciences [25], has increased significantly.

In summary, this study’s aims were to obtain more evidence of HBOT research trends from the information in the database through a visual bibliometric method, so as to point out the shortcomings of past research and the directions of future research.

## 2. Data Information and Methodology

WOS platforms make up one of the largest scientific citation index databases on the world. The comprehensive academic information resources consist of six online indexes that include more than 8700 core academic journals in various fields such as medical science, biomedicine, engineering, humanities and social sciences. In the data mining process, digital system parameters were searched from WOS citation indexes (SCI, SSCI, AHCI and ESCI). The field retrieval of information was “title” and “topic”. The digital system parameter search term setting of “hyperbaric oxygen therapy” was used for this study. The document type was limited to “article”. Articles were excluded if their primary focus was not HBOT, even if the topic was peripherally discussed. Furthermore, publications such as meetings and commentaries, letters to the editor, editorials and abstracts were excluded. Since this study is concerned with the development trend of HBOT in recent years, 1999–2021 is used as the time interval for analysis. According to the stated inclusion and exclusion principles, 5944 articles on HBOT were obtained.

VOSviewer tools (version 1.6.15) were used in this study. They were developed by the Centre for Science and Technology Studies at the University of Leiden. VOSviewer is a free access visualization tool that can quickly render numerical data visually to make published content stand out. Additionally, VOSviewer is able to recognize any patterns in large bibliometric datasets, including Europe PMC, Microsoft Academic, Open Citations, Crossref and PubMed [26]. This mapping technique has been developed in a variety of fields, and its capabilities are widely used to investigate major research trends and subfields. In particular, the citation network and keyword co-occurrence analysis are analytical advantages.

In this study, we assessed the various functional options of the system and accurately analysed the growth of publications and research hotspots related to HBOT. The results of this analysis will contribute to the development of HBOT. Details on the steps for this study are shown in Figure 1.

The VOSviewer clustering technique is based on the connections between its cluster function (e.g., bibliographic coupling and co-authorship relations) and the strength among objects of interest (the body of scientific literature) [27]. This visualization technique constructs a similarity matrix from a co-occurrence matrix using a similarity measure with the strength of association [28], and constructs a two-dimensional view wherein the position of objects is refined and the distance among different pairs of objects is calculated accurately based on their similarity. The VOSviewer mapping analysis is designed to minimize a weighted sum of squared Euclidean distances among all pairs of elements through an enhancement to the process [29]. It allows examination through three views (label view, density view and cluster density view). Figure 2 shows the setups 1 to 5 of the VOSviewer software technical drawing application used in this study.

## 3. Results of Visualization Analysis

In this paper, we determined the most influential journals, authors and countries publishing HBOT research. Additionally, the visualization map (citation, co-citation and co-occurrence) was constructed in this study, and based on this, the theoretical foundation for the visualization platform research was established.

### 3.1. Bibliometric Analysis of the Collaboration Network

Figure 3 shows the relationships in the collaboration network. The significance of this collaboration network, in which nodes correspond to countries, is that larger nodes indicate that more publications have been contributed by the corresponding country. The colours of the circles indicate the weight of the citation item, with warm colours being more important than cool colours. This map highlights the extent to which international research collaborations are linked. A total of eight clusters are visible. The country with the greatest number of publications is the United States (cluster 1, 11 items, red), and others include England, Mexico, Sweden, Norway, Denmark, France and Scotland. These countries make up the cluster with the highest number of publications. This is followed by a smaller cluster of countries, including Israel, Turkey, Poland, New Zealand and Portugal (cluster 2, 8 items, green). These countries have also established good cooperative relations with each other. In cluster 3 (5 items, blue), there is another cluster of countries, including Greece, Croatia, Italy, the Czech Republic and Germany. In cluster 4 (5 items, khaki), the countries include Brazil, Canada, Finland, Serbia and South Korea. In cluster 5 (5 items, lavender) they consist of Austria, Indonesia, Japan and Spain. In cluster 6 (3 items, baby blue), the constituent countries cover China, Iran and Taiwan. Overall, a total of five Western countries were in the top 10 for this indicator. The fact that there are few connections between cluster 7 (India and Russia, orange) and cluster 8 (Slovenia and Switzerland, wheat) implies that there is a weak cooperative relationship with other countries in the HBOT academic field.

Figure 4 shows the number of publications for each country or region. A total of 1637 articles were published in journals from the USA, which has the highest concentration of publications. The countries with the second-highest number of research publications are China and Turkey; both countries have 416 published articles. In terms of the number of articles, however, the research results published by China and Turkey are still far fewer than the research output of the USA. Other countries that have a high number of publications are as follows: Japan (*n* = 238), the UK (*n* = 186), Australia (*n* = 176), Taiwan (*n* = 176), Israel (*n* = 156), Germany (*n* = 154) and Italy (*n* = 136).

In addition, the USA has a link strength of 12,741 with 30,669 citations; the USA ranks significantly higher than other countries in total link strength, which is expected because of their number of publications. China and Turkey are second to the United States in total link strength; China’s link strength is at 3860, with 4568 citations, while Turkey’s link strength is 3679 with 3872 citations. Interestingly, the scientific research production related to HBOT is concentrated in the northern hemisphere, with little representation from the southern hemisphere, which includes Indonesia and South Africa.

### 3.2. Disciplinary Collaboration based on Co-Authorship

Figure 5 shows the interrelationships of authors’ intensity, which changes yearly and is illustrated by a gradient of blue to yellow. A total of 45,246 researchers were included in this analysis. The most striking name, shown in a large yellow area, is S. R. Thom. His research has received widespread attention in hyperbaric oxygen therapy for acute carbon monoxide poisoning. The advantage of HBOT treatment is not only to increase the dissolved oxygen content in the blood, but also to help protect heart and brain tissue in Thom’s study [30]. R. E. Marx is another important researcher in HBOT research. He found that HBOT combined with aggressive surgery is a progressive strategy for the treatment of osteoradionecrosis [31]. The results from the analysis indicate that the sustainable growth of citations published by authors has a far-reaching impact on the future development and direction of HBOT research.

The results of the author co-citation analysis and the investigation of the interrelationships among individual authors in the academic field are presented in Figure 6. S. R. Thom is the most productive contact author; his articles have a total of 1157 citations and a link strength of 14,264. Other well-known scholars in HBOT research include R. E. Marx and T. K. Hunt [32]; Marx’s articles have 819 citations and a total link strength of 6938. Meanwhile, Hunt’s articles have 436 citations and a total link strength of 6727. Figure 6 lists the most cited researchers by co-citation frequency. These highly co-cited authors are recognized as knowledge hierarchy pioneers who have developed strategies to improve HBOT.

### 3.3. Visualization Analysis of the Journals That Published HBOT Research

Figure 7 shows the leading global journals that have published scientific articles about HBOT. In the overlay map, the colourful circles each correspond to a scientific journal. The size of the circle represents the journal’s citation weight. This visualization map identifies eight different clusters based on this distribution. The three big clusters on this map are green, blue and red. The green cluster (Cluster 1) consists of members such as Undersea and Hyperbaric Medicine, the Archive of Surgery—Chicago and the New England Journal of Medicine. The red cluster (Cluster 2) has citations from the Journal of Biological Chemistry, Journal of Surgical Research, Critical Care Medicine and others. The formation of the other clusters is based on the same structure. Finally, the purple cluster includes The Lancet, the Journal of Applied Physiology and the American Journal of Physical Medicine & Rehabilitation. These listed journals have the maximum number of co-citations, and it is beneficial to grasp the journals’ contribution to medicine and competencies.

The results of this study reveal information from most publications in the WOS. The findings show that the citation analysis of journals reflects the research domains of HBOT (Figure 8). Articles published in *Undersea and Hyperbaric Medicine* have been co-cited 2843 times, and have a total link strength of 105,698. Other journals with high citations and overall link strength include the *Journal of Applied Physiology* and *Brain Research*; the *Journal of Applied Physiology* has a total of 1728 citations and a total link strength of 59,549, while *Brain Research* has a total of 1443 citations and a total link strength of 59,541. It is noteworthy that these well-known journals are focused on the clinical applications on orthopaedics, plastic surgery, neurology, rheumatology and neurosurgery. This shows that these clinically oriented journals can influence future research directions.

### 3.4. Visualization Analysis of Dynamic Changes in Co-Occurrence

Structural analysis of keywords helps to identify important research topics in the field. Visualization co-occurrence keyword analysis is an efficient tool providing insight into the most popular topics in publications in a specific research field and how their frequency varies with time [33].

Results show that from the bibliographic database of 5594 articles with the minimum occurrence pre-set at 10, there are 8744 keywords in 433 articles that meet the threshold. The total link strength of each keyword was calculated, and is shown in Figure 9. The HBOT keywords with the greatest link strength stand out. We visualized in colour the most searched keywords according to search frequency. The red cluster corresponds to the terms searched most frequently (83 items) and the keywords represented include hyperbaric oxygen, oxidative stress and brain. The green cluster includes the second-most frequently searched terms (78 items) and the keywords represented include ischemia, expression and stroke. The third cluster is blue (71 items), which includes keywords such as hyperbaric oxygen therapy, therapy and management. The fourth cluster is yellow (59 items) and the keywords represented include radiotherapy, injury and complication. Finally, the keywords with the lowest search frequency make up the purple cluster (44 items), and the keywords represented include traumatic brain injury, hypoxia and pressure.

The findings show that HBOT research focuses on its mechanisms, indications and treatment risks. These data can help researchers speculate on the future directions of HBOT research. In addition, the management of HBOT remains the most important research topic.

As shown in the overlay visualization in Figure 10, we present the results of the overlay visualization that classifies the described network connections from 2006 to 2016. The different colours and sizes of the circles are positively correlated with the frequency of occurrence in the papers. In other words, the colour of an item is determined by matching the colour value of an item with the colour value (year) in an overlay colours file. It can be observed that the keywords hyperbaric oxygen, brain and embolism had a higher frequency of co-occurrence in HBOT research between 2006 and 2009. During 2010–2013, the keywords associated with wound healing, specifically complications, therapy and embolism, appeared more frequently. From 2014 to 2016, the keywords hyperbaric oxygen, management and efficacy had high frequencies. Finally, after 2017, the keywords sudden hearing loss, management, symptoms and combination therapy received more research attention.

Co-occurrence network analysis was used to examine the relationships in and structure of HBOT research (Figure 11).

It can be observed that the keyword with the highest co-occurrence frequency density and total link strength is “hyperbaric oxygen”, with values of 912 and 4459, respectively. The “treatment” of co-occurrence frequency is 672 and the total link strength is 3406. Lastly, the “hyperbaric oxygen therapy” of co-occurrence frequency is 635, and its total link strength is 2421. The remaining keywords have lower co-occurrence frequency and total link strength.

This also subtly shows that the keywords showed a research concentration on the risks and benefits of HBOT. This can be partly attributed to attention from the academic community, and it is still an important subtopic in HBOT research.

## 4. Discussion

According to the results of the visualization analysis of HBOT publications, we propose the following implications for HBOT research and trends in coming years.

### 4.1. Identified and Engaged: Academic Research Performance and Collaborative Influence

Bibliometric analysis is based on constructing graphical databases with various citation metrics. It is frequently used in the field of library and information science [34]. The global use of the Internet has eliminated the limits of physical distance between countries, making it easier for people to cooperate with each other; this has contributed to changes in citation analysis. Due to international scientific collaborations, there have also been recent changes in the traditional world-systems theory that can reasonably explain new situations. This has been called a knowledge movement [35]. This trend emphasizes intellectual activity and opportunities across international borders, and allows anyone to practice and challenge it. The USA is well-known for leading research in this field, as it has a powerful international collaborative network with the European region, China, Turkey, Japan and Australia. This indicates that countries must cooperate and that academic intelligence must be strengthened. Simply put, international academic cooperation is recognized by experts as the most cost-effective and credible solution available to advance knowledge on HBOT.

The UHMS is an international non-profit organization that is the primary source of scientific information for hyperbaric medicine in relation to physiology and diving [36]. In addition, the UHMS also developed several evidence-based educational programs around the world specific to aspects of HBOT in biology, medicine, physical science, therapeutics, monitoring, specific technical projects and applied research. The UHMS thus plays an important role in the progress of HBOT research.

Based on the number of journals and authors mentioned above, there is strong competition globally to contribute to HBOT. This study confirms the use of bibliometric analysis to examine the status of international academic cooperation and identify core research groups. The USA is a core country with a dynamic comparative advantage; it also has technological spillover, and is a leader in HBOT research and innovation.

### 4.2. A Powerful Structural Graph and Visualization Analysis Model Can Provide New Information on HBOT

The article that received the most citations has also exerted a positive and profound influence on current research [37]. As in the analysis of the aforementioned co-authorship results, three researchers whose work is highly cited in the HBOT field are worthy of attention. S. R. Thom focused on the effects of high-pressure gases in his early research [38]. A total of 33 publications included the evaluation and treatment of patients with carbon monoxide poisoning [39]. He next studied, in great detail, oxidative stress responses and their pathological and toxicological implications, as well as the development of therapeutic modalities of HBOT. These studies showed that HBOT helped prevent oxidative damage. There were a total of 17 studies in this area, covering topics such as wound healing and the role of stem cells [40]. After that, he employed these theories (fundamental cell biology and biochemical investigations) to examine whether exposure to high partial pressures of oxygen increased reactive species-derived clinical responses and improved the outcomes of some reperfusion injuries [41]. In sum, Thom’s ideas on HBOT have created an intellectual tower. His publications have been highly cited over the last thirty years. His research activity and scholarly programs can be considered to have impacted the course of other scientists’ careers.

R. E. Marx is another researcher who has made important contributions in HBOT. He pioneered new concepts and treatments for oral pathologies, and made valuable contributions to the use of hyperbaric oxygen after radiation therapy. He indicated the benefits of hyperbaric oxygen therapy, including vascular improvement, maintenance of intact mucosa, expedited healing of osteoradionecrosis and skin over all bone and better tolerance to surgical wounding prior to oral surgery [42]. In addition, his extensive research fields have also opened new horizons in HBOT research, including the relationship between oxygen dose to angiogenesis induction in irradiated tissue, a new concept of osteoradionecrosis treatment [43] and the application of hyperbaric oxygen therapy in bony reconstruction of the irradiated tissue-deficient patient [44]. The aforementioned studies illustrate his research results in oral and maxillofacial surgery.

The next most important contributor on HBOT is T. K. Hunt. He is dedicated to curing age-related diseases through biotechnology. T. K. Hunt has long been immersed in the development of trauma services and the cell biology of wound healing [45]. In exploring strategies to enhance angiogenesis, he injected hyperbaric oxygen into tissue to aid in the healing of surgical wounds [46]. Others include the role of oxygen in the repair process [47], the treatment of osteoradionecrosis with hyperbaric oxygen [48] and the repair of the gut mucosa [49]. One of his important contributions was the simple application of oxygen to increase vasodilation. This treatment can significantly reduce infection and improve healing ability of tissue.

The analysis of journals that have published papers on HBOT shows publishers that engage with similar subjects. The *Undersea and Hyperbaric Medicine* journal is the most important in the HBOT field, having received 2843 citations, and has a co-citation strength of 105,698. This journal is receptive to new scientific information on this topic and has great publication strength, which has contributed to conceptual research and opinions on future research directions for HBOT. The *Journal of Applied Physiology* focuses on the theories, methods and applications of scientific concepts and publishes techniques in physiology. *Brain Research* is devoted to fundamental research on the brain and neuroscience. It publishes articles on nervous system structure and function, diagnoses of brain injuries and experiments and clinical trials for diseases that are the relevant for HBOT and of general interest to the international community of neuroscientists. Other journals, such as the *Stroke* and *The Lancet*, also play an important role in HBOT research.

These topics not only provide new scientific perspectives, theoretical foundations and clinical applications in hyperbaric medicine, but also seek to address urgent topics in society, such as human health and major environmental events. Notably, *Undersea and Hyperbaric Medicine* is the most frequently cited journal in our analysis (105,698 co-citation strength). Although some journals do not have many publications, they have a very high number of co-citations. It is also worth noting that based on the co-journal analysis, the *Unersea and Hyperbaric Medicine* journal predominates, especially in terms of the exceptional quality of its research articles. It makes a substantial scientific contribution and has a far more positive impact and the greatest effect on academic HBOT research compared to any other journal. In the classification of the top five journals related to HBOT, relevant subjects included plastic surgery, neurological medicine, physiology and neurosurgery. Higher co-citations and effects on the modern-day health system show that journals add strength and value to research that positively impacts the lives of people.

We summarize our arguments as follows. The collaborative network of authors and journals shown in the bibliometric analysis contributes to highlighting lesser-known disciplines and identifies potential competitors. Capable scientists are necessary for improving these theories, monitoring the future of HBOT, contributing to solutions to recent global issues and combining creative imagination with true scholarship.

### 4.3. The Call to Action: High-Quality Research Is Crucial for Health Systems and Can Improve Health and Impact People’s Lives

Co-keyword analysis can effectively measure the relatedness of topics and quantitative knowledge structure [50]. It is an effective method to identify hot topics [51]. The characteristic visualization network was obtained by analyzing Pearson’s correlation matrix [52]. The results of bibliometric analysis can help the researchers to think from the perspective of macroscopic and geographic boundaries beyond the field [53]. It plays an important role in exploring the value of HBOT.

From the analysis results of important keywords, hyperbaric oxygen therapy, oxidative stress and expression are the research topics that attract attention. The bibliometric analysis highlighted some of the beneficial effects of HBOT. The data that indicate the therapeutic effects of hyperoxia have been demonstrated often, but this research topic still attracts a wide range of public opinions. For example, HBOT is currently used off-label to treat COVID-19 pneumonia patients. An overly active immune response generating inflammation is a direct cause of the pathogenesis of the SARS-CoV2 virus. These studies have shown that HBOT has a reliable threshold value on oxygen absorption and its delivery into blood vessels and tissue, thereby increasing oxygen saturation [54]. Recent controversial issues on the use of HBOT include sudden hearing loss [55] and medical science aesthetics [56]. These results from studies on HBOT have high citation frequencies, and reflect the attractiveness of these topics to scholars in other fields.

In addition, among the indications not covered by health insurance, there is empirical medical evidence of their auxiliary effects of sudden deafness, skin flap transplantation [57], poorly healed wounds [58], acute stroke [59] and fibromyalgia [60]. Therefore, the quality of HBOT should be studied in a standardized and measurable way to further study the benefits of its use as an adjuvant therapy in other diseases. It is important to note that there are differences in the care channels used in the studies reviewed and further exploration into the appropriateness of whether these treatment options should be funded by the healthcare system is necessary.

At present, governments, as medical advocates and policymakers, should formulate future guidelines and academic research directions for HBOT, seek to find common ground with other countries and reach agreements to resolve issues and conflicts of common concern. Although considerable progress has been made in this field in recent years, more research and evidence are needed to establish the status of HBOT in 21st century medicine. More importantly, a clearer strategic direction and broader development space for HBOT should be formulated to stimulate the development of the medical field and improve the quality of medical care.

## 5. Conclusions

Broadly, there is growing evidence that HBOT substantially contributes to the treatment of disease, both in primary management and as an adjuvant. Therefore, the focus of future research could be determining whether HBOT has benefits in the treatment of diseases such sudden deafness, Parkinson’s syndrome, dementia, cancer and other diseases, and even currently prevalent diseases such as COVID-19. This study confirms the mechanisms underlying HBOT and provides indications for treatment through statistics and analysis of evidence from extensive literature databases and information platforms. In this way, it provides a reference for the payment of medical insurance in various countries and application of limited medical resources to the most appropriate and necessary medical behaviours. Finally, although some interesting results were obtained through the bibliometric analysis and visualization of publications related to HBOT, there were also several research limitations to this process. The data sources for the study were archives downloaded from the WOS SSCI and Science Citation Index Extended databases. More than 99% of these articles are written in English, which leads to an underestimation of the research literature and authors in other languages. In the future, researchers can use other software (e.g., BibExcel, CiteSpace and Pajek) for classification, comparison and analysis. This approach can make bibliometric analysis more objective and efficient. There are also opportunities for cross-certification in terms of the proofreading and interpretation of academic progress. The results of this study provide clinically correct uses of HBOT, can provide indications and can enable medical benefits to be applied to patients who are most in need.

## Figures and Tables

**Figure 1 ijerph-19-07866-f001:**
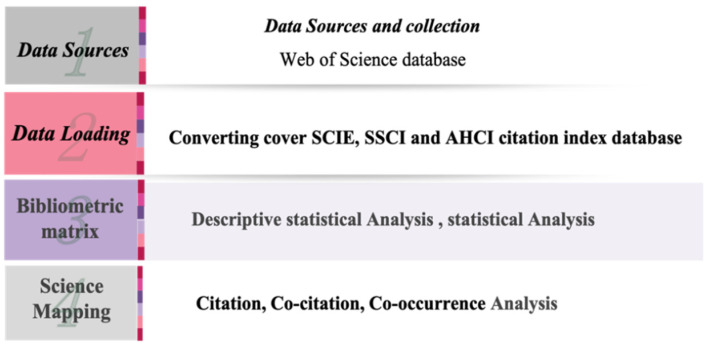
Visualization analysis and science-mapping workflow.

**Figure 2 ijerph-19-07866-f002:**
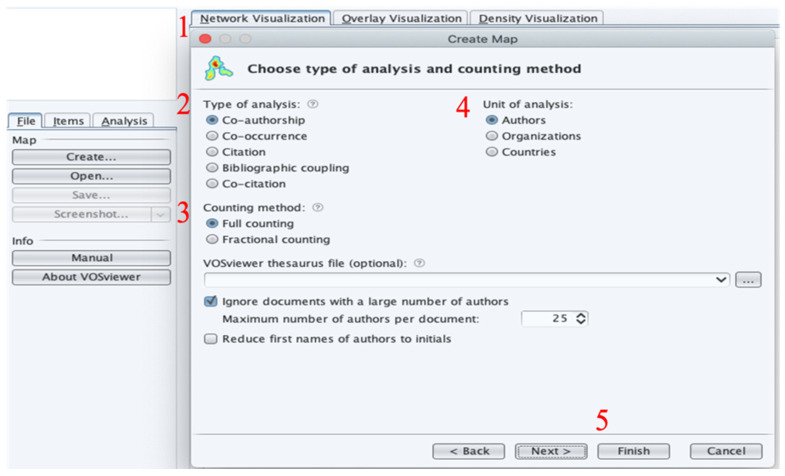
VOSviewer science-mapping workflow.

**Figure 3 ijerph-19-07866-f003:**
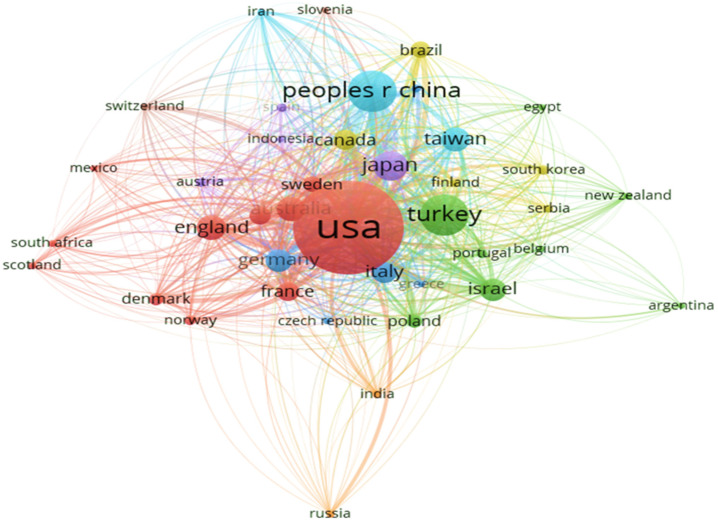
VOSviewer network visualization of research collaborations among countries.

**Figure 4 ijerph-19-07866-f004:**
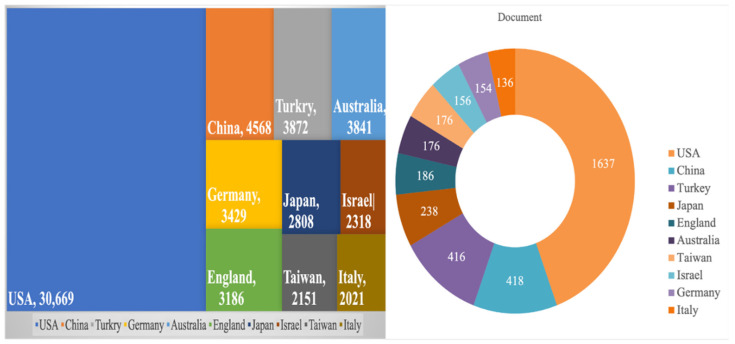
Distribution of the number of publications in each country.

**Figure 5 ijerph-19-07866-f005:**
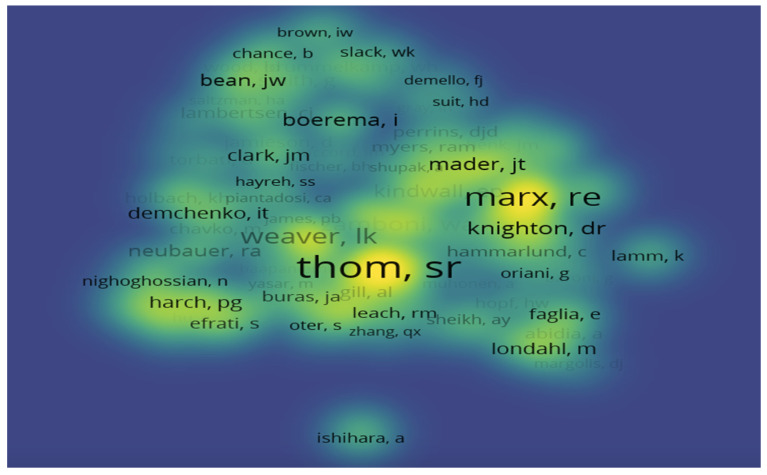
Visualization of co-citation density.

**Figure 6 ijerph-19-07866-f006:**
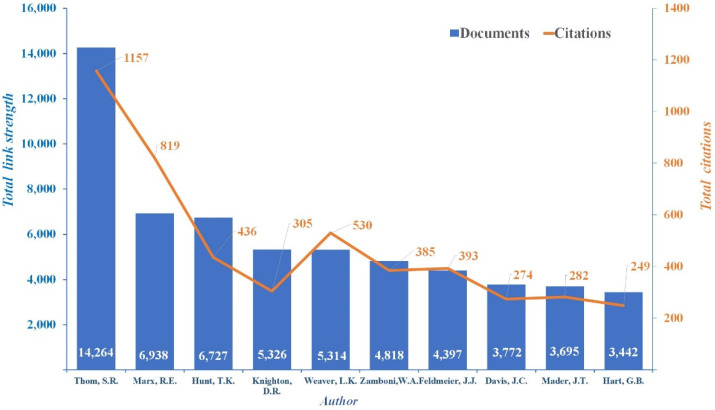
Bibliometric analysis of the most active co-authors’ relationships.

**Figure 7 ijerph-19-07866-f007:**
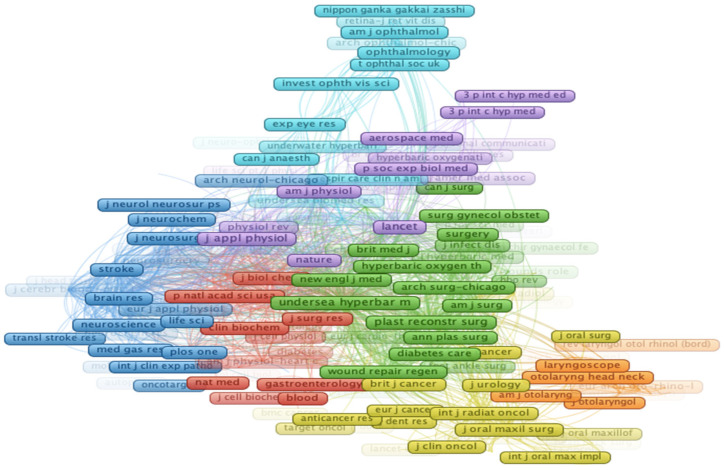
Overlay visualization of the journals.

**Figure 8 ijerph-19-07866-f008:**
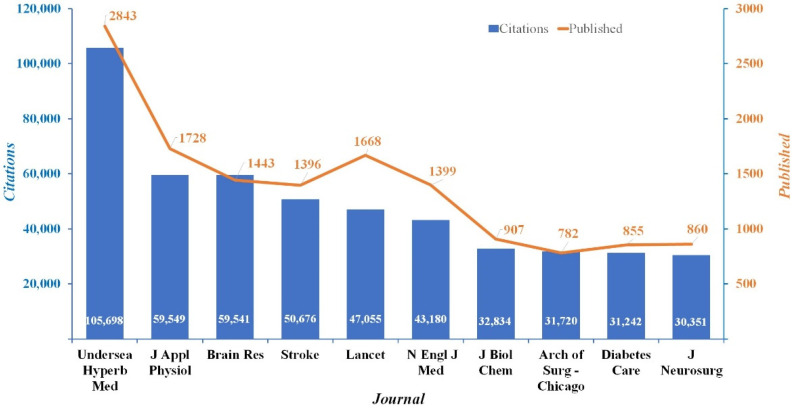
Distribution of journals publishing studies related to HBOT research.

**Figure 9 ijerph-19-07866-f009:**
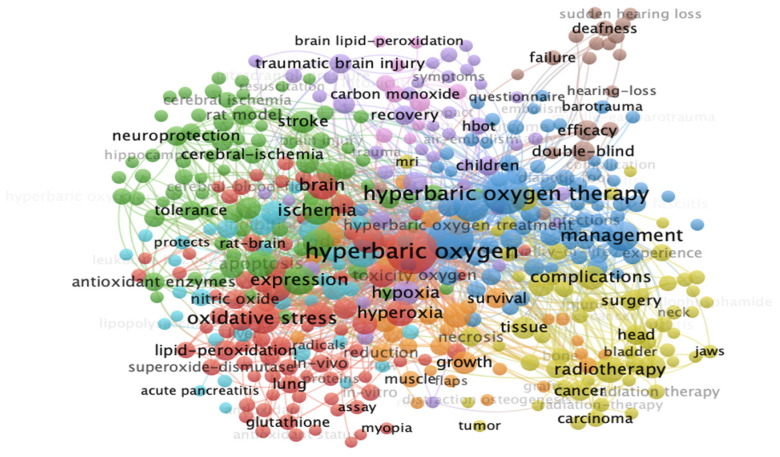
Network visualization of keyword co-occurrence.

**Figure 10 ijerph-19-07866-f010:**
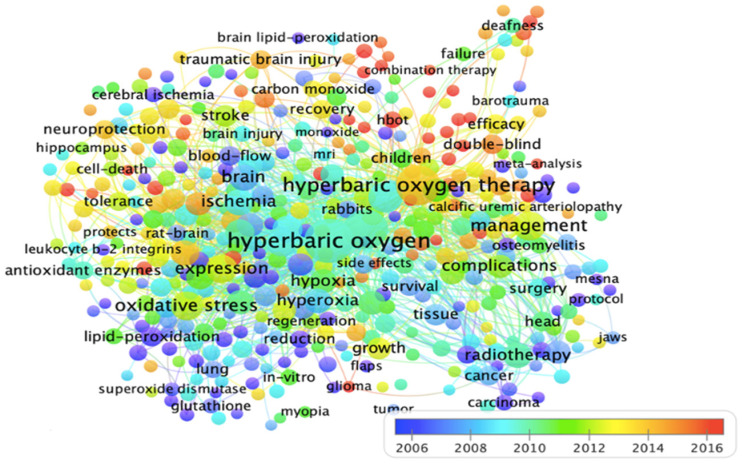
Overlay visualization of keyword co-occurrence.

**Figure 11 ijerph-19-07866-f011:**
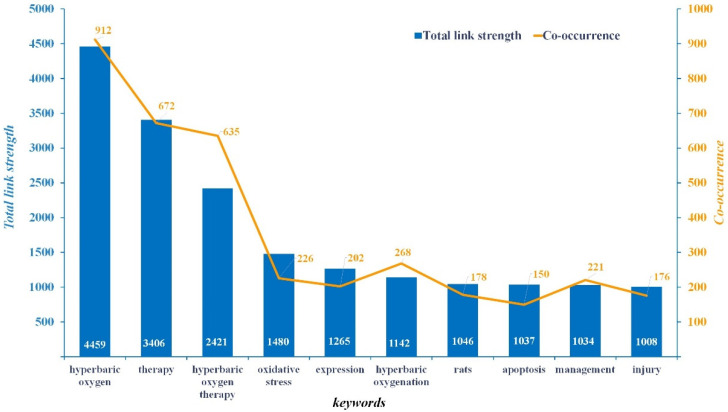
Keywords with the strongest occurrences and total link strength in HBOT research.

## Data Availability

Raw and processed data are available upon request to the corresponding author.

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
