# Peer review of "Visualization and Bibliometric Analysis of Research Trends on Hyperbaric Oxygen Therapy"

_ijerph, 2022, doi:10.3390/ijerph19137866_

Round 1

Reviewer 1 Report

Synopsis

This study applied bibliometrics to assess bibliographic data in the expansive integrated web platform (WOS) using VOSviewer.  Visualization analyses identified concentrations of international research collaboration, country or region of origin of publications, leading individual researchers, co-citation density, publication journals, and keyword searches.   A limitation is the use of databases that are largely limited to publications published in English, thereby overlooking studies in non-English languages and scientific journals.  But this is determined by the availability of appropriate databases.

Critique

This is a well-crafted review of visualization and bibliometric analyses of hyperbaric oxygen therapy (HBOT) research.  This is an outstanding contribution to the HBOT literature.

Bibliometrics is often overlooked by more hardcore basic and clinical science researchers.  However, it can be an invaluable tool for identifying trends in HBOT research that can highlight emerging areas of research leading to new clinical indications. 

Author Response

Once again, thank you very much for the comments and suggestions on our revised manuscript. We are grateful to be given the opportunity to submit our revised manuscript, and to respond to some final-stage concerns. We have carefully considered the reviewers’ suggestions and advice, and made further improvements to the manuscript.  Responses to each individual comment are contained in the below.

In addition, in order to make the English usage of the manuscript meet the requirements of academic journals, the manuscript has been edited by the English editing service provided by MDPI (English-Editing-Certificate-45269). The use of English in the manuscript should meet academic requirements.

Kind regards, 
The Authors 

Comments:

Synopsis
This study applied bibliometrics to assess bibliographic data in the expansive integrated web platform (WOS) using VOSviewer.  Visualization analyses identified concentrations of international research collaboration, country or region of origin of publications, leading individual researchers, co-citation density, publication journals, and keyword searches.   A limitation is the use of databases that are largely limited to publications published in English, thereby overlooking studies in non-English languages and scientific journals.  But this is determined by the availability of appropriate databases.

Respones:

Thanks. Due to the language limitation of the documents included in the database, this selection condition may miss some important studies in non-English languages. This is also one of the limitations of this study. However, the WOS database does already cover most of the important literature written in English. Therefore, the findings of this study still make a positive contribution to the understanding of HBOT. The limitations of this research have been expressed in the manuscript. Please see line 415-422.

Comments:

Critique
This is a well-crafted review of visualization and bibliometric analyses of hyperbaric oxygen therapy (HBOT) research.  This is an outstanding contribution to the HBOT literature.

Bibliometrics is often overlooked by more hardcore basic and clinical science researchers.  However, it can be an invaluable tool for identifying trends in HBOT research that can highlight emerging areas of research leading to new clinical indications.

Respones:

Thanks to the reviewers for the positive affirmation of this manuscript.

Reviewer 2 Report

I've given this manuscript a first glance, but have not performed an in-depth review because I believe the manuscript requires major modifications before it can be further considered.

The basic idea to report on bibliometrics and research collaborations in HBOT is interesting. This could be done in a short and consice manner, highlighting only the important points and refraining from too many details, elaborations on the position of HBOT in the scientific field and matters of reimbursement, et cetera. Also, the English language is unnecessarily woolly.

I will continue with a list of suggestions, but I wil stress that this list is far  from complete, as I've not gone through the whole manuscript in detail.

  • Headers do not match the requirement of the journal (e.g. nr 3 should just be titled "Results")
  • Introduction does not read fluently, is too long, and contains too many elaborations. I suggest to start with a short overview of the therapeutic uses of HBOT (without mentioning random applications such as improving fatigue after exercise, which is not an accepted indication). Please stick to the UHMS list of indications. Then continue to say that we are lacking a compehensive overview of HBOT bibliometrics and that having this could provide insight into previous research and thereby direction for the future. Do not mention methodological issues in the introduction (the part that starts with "The Web of Science (WOS) is one of..." is methodology).
  • Please be more clear about the goal of the study, as it currently stands this goal is hidden in a lot of noise.
  • Refrain from words such as "famous" (first line of paragraph 2.) and "extremely easy-to-use" (first line of page 3). Remain factual and scientific, remove all empty sentences/clauses.
  • Why were studies prior to 1999 not included? This article provides a very good opportunity to provide an overview of all of HBOT research, not only >1999.
  • Why is there mentioning of the weak link between Serbia and Egypt? Just an example? It seems very random.
  • Two major articles are highlighted in particular, the articles by Marx and Thom. It doesn't make very much sense to me to pick these two out of the vast amount of HBOT papers.
  • When it comes to the results, the major points are covered in a lot of noise, which causes distraction. Show only the most important results.
  • Discussion: stick to the goal of the paper, discuss the bibliometric findings and draw some conclusions from these. I do not understand why there is a major paragraph on medicolegal and reimbursement issues.
  • "The call to action: High-quality research is crucial for health systems, can create an unparalleled environment, and impact people’s lives" what is an unparalleled environment?? This is an example of an empty clause / sentence.

I believe the purpose of the paper is sound, but it requires a lot of revision to be considered for publication. Stick to the main purpose, remove all unnecessary elaborations (by this I mean superfluous paragraphs that do not contribute towards the main goal, as well as empty/woolly sentences/clauses). Use only factual and scientific wording

Author Response

Once again, thank you very much for the comments and suggestions on our revised manuscript. We are grateful to be given the opportunity to submit our revised manuscript, and to respond to some final-stage concerns. We have carefully considered the reviewers’ suggestions and advice, and made further improvements to the manuscript.  Responses to each individual comment are contained in the below.

In addition, in order to make the English usage of the manuscript meet the requirements of academic journals, the manuscript has been edited by the English editing service provided by MDPI (English-Editing-Certificate-45269). The use of English in the manuscript should meet academic requirements.

Comments:

I've given this manuscript a first glance, but have not performed an in-depth review because I believe the manuscript requires major modifications before it can be further considered.

Response:

Thanks for the reviewer's suggestion. We followed the reviewer's suggestion to revise this manuscript.

Comments:

The basic idea to report on bibliometrics and research collaborations in HBOT is interesting. This could be done in a short and consice manner, highlighting only the important points and refraining from too many details, elaborations on the position of HBOT in the scientific field and matters of reimbursement, et cetera. Also, the English language is unnecessarily woolly.

Response:

Thanks for the reviewer's suggestion. We have attempted to remove unnecessary narratives to highlight important features of the manuscript. In addition, the English usage of the manuscript has been moderately corrected and edited by IJERPH's English editing service (English-Editing-Certificate-45269). The English use of the manuscript should comply with the requirements of IJERPH.

Comments:

I will continue with a list of suggestions, but I will stress that this list is far from complete, as I've not gone through the whole manuscript in detail.

Response:

Thanks for the reviewer's suggestion. We followed the reviewer's suggestion to revise this manuscript.

Comments:

Headers do not match the requirement of the journal (e.g. nr 3 should just be titled "Results")

Response:

Thanks for the reviewer's suggestion. We've adjusted the bug in the title. Please see line 109.

Comments:

Introduction does not read fluently, is too long, and contains too many elaborations. I suggest to start with a short overview of the therapeutic uses of HBOT (without mentioning random applications such as improving fatigue after exercise, which is not an accepted indication). Please stick to the UHMS list of indications. Then continue to say that we are lacking a compehensive overview of HBOT bibliometrics and that having this could provide insight into previous research and thereby direction for the future. Do not mention methodological issues in the introduction (the part that starts with "The Web of Science (WOS) is one of..." is methodology).

Response:

Thanks for the reviewer's suggestion. We have revised the description of the "Introduction" section following the reviewer's suggestion. Please see line 52-61.

Comments:

Please be more clear about the goal of the study, as it currently stands this goal is hidden in a lot of noise.

Response:

Thanks for the reviewer's suggestion. We have revised unnecessary text in the manuscript. Such treatment can make the purpose of this study clearer. Please see line 62-64.

Comments:

Refrain from words such as "famous" (first line of paragraph 2.) and "extremely easy-to-use" (first line of page 3). Remain factual and scientific, remove all empty sentences/clauses.

Response:

Thanks for the reviewer's suggestion. We've removed unnecessary words. Please see line 66 and line 79-82.

Comments:

Why were studies prior to 1999 not included? This article provides a very good opportunity to provide an overview of all of HBOT research, not only >1999.

Response:

Thanks for the reviewer's suggestion. Although HBOT has been used in medicine as early as 1887, the most important research advances have been made in recent years. The purpose of bibliometric analysis is to provide instructive research guidelines for future research. Therefore, the interval of data selection is based on the last 20 years in this study. Relevant instructions have been described in the manuscript. Please see line 76-77.

Comments:

Why is there mentioning of the weak link between Serbia and Egypt? Just an example? It seems very random.

Response:

Thanks for the reviewer's suggestion. We rearranged the relationships of collaboration network between countries. Such revisions should make the results clearer. Please see line 120-134.

Comments:

Two major articles are highlighted in particular, the articles by Marx and Thom. It doesn't make very much sense to me to pick these two out of the vast amount of HBOT papers.

Response:

Thanks for the reviewer's suggestion. We revised the way the manuscript is presented. For important HBOT investigators, we adopted a strategy of comprehensively reviewing the overall contribution of that author. Please see line 160-168.

Comments:

When it comes to the results, the major points are covered in a lot of noise, which causes distraction. Show only the most important results.

Response:

Thanks for the reviewer's suggestion. We revised the expression in the manuscript. Please review the content of the "Results of Visualization Analysis" section again.

Comments:

Discussion: stick to the goal of the paper, discuss the bibliometric findings and draw some conclusions from these. I do not understand why there is a major paragraph on medicolegal and reimbursement issues.

Response:

Thanks for the reviewer's suggestion. We revised the content of "4.1" and "4.2" and removed content that was not relevant to the research results. Such a statement can make the manuscript more focused. Please review the content of these subsections again.

Comments:

"The call to action: High-quality research is crucial for health systems, can create an unparalleled environment, and impact people’s lives" what is an unparalleled environment?? This is an example of an empty clause / sentence.

Response:

Thanks for the reviewer's suggestion. We have revised the title for "4.3" (Please see line 367-368). Furthermore, in this subsection, we also remove unnecessary statements. Please review the content of this subsection again.

Comments:

I believe the purpose of the paper is sound, but it requires a lot of revision to be considered for publication. Stick to the main purpose, remove all unnecessary elaborations (by this I mean superfluous paragraphs that do not contribute towards the main goal, as well as empty/woolly sentences/clauses). Use only factual and scientific wording.

Response:

Thanks for the reviewer's suggestion. We followed the reviewer's suggestion to revise this manuscript.

Reviewer 3 Report

This work is very interesting. There are no studies analyzing the works done on HBOT as a topic. 

The authors analysis is informative and useful in the reader's point of view. 

I have a couple of suggestions for improving the manuscript.  

1. Please provide detailed information on "VOSviewer."   It is mentioned as a free software package, but where it is provided is unclear: company, version, etc. 

2. for most of the figures please choose the colors that will improve the readability. 

3. For figure 8, please provide the index (yellow bar and green bar)

Overall, interesting and good descriptive work. 

Author Response

Once again, thank you very much for the comments and suggestions on our revised manuscript. We are grateful to be given the opportunity to submit our revised manuscript, and to respond to some final-stage concerns. We have carefully considered the reviewers’ suggestions and advice, and made further improvements to the manuscript.  Responses to each individual comment are contained in the below.

In addition, in order to make the English usage of the manuscript meet the requirements of academic journals, the manuscript has been edited by the English editing service provided by MDPI (English-Editing-Certificate-45269). The use of English in the manuscript should meet academic requirements.

Comments:

This work is very interesting. There are no studies analyzing the works done on HBOT as a topic. 

The authors analysis is informative and useful in the reader's point of view. 

I have a couple of suggestions for improving the manuscript.

Response:

Thanks to the reviewers for the positive affirmation of this manuscript.

Comments:

1. Please provide detailed information on "VOSviewer."   It is mentioned as a free software package, but where it is provided is unclear: company, version, etc.

Response:

Thanks for the reviewer's suggestion. We have added VOSviewer version and publisher information. Please see line 79-80.

Comments:

2. for most of the figures please choose the colors that will improve the readability.

Response:

In Figures 3, 6, 8, and 11, these figures have been corrected. Such processing can increase the degree of recognition. Additionally, in Figures 5, 7, 9, and 10, these figures are based on the image stacking function of the VOSviewer analysis system. They will have color differences due to the number of citation strengths. After considering the authenticity of the analysis, we have presented the clearest image based on the system color correction function.

Comments:

3. For figure 8, please provide the index (yellow bar and green bar).

Response:

Thanks for the reviewer's suggestion. We already have finished the relationship between the position of the specified index number in fig 8.

This manuscript is a resubmission of an earlier submission. The following is a list of the peer review reports and author responses from that submission.